# A Review on Daphnane-Type Diterpenoids and Their Bioactive Studies

**DOI:** 10.3390/molecules24091842

**Published:** 2019-05-13

**Authors:** Yue-Xian Jin, Lei-Ling Shi, Da-Peng Zhang, Hong-Yan Wei, Yuan Si, Guo-Xu Ma, Jing Zhang

**Affiliations:** 1College of Chinese Medicine Material, Jilin Agricultural University, Changchun 130118, China; jyx0107@126.com (Y.-X.J.); siy2826405514@163.com (Y.S.); 2Institute of Medicinal Plant Development, Chinese Academy of Medical Sciences and PekingUnion Medical College, Beijing 100193, China; 3Xinjiang Institute of Chinese and Ethnic Medicine, Urumqi 830002, China; shileiling@sina.com (L.-L.S.); whywlmq@sina.com (H.-Y.W.); 4College of Life Science and Technology, Xinjiang University, Urumqi 830046, China; zdp88888@yeah.net

**Keywords:** daphnane, diterpenoid, cytotoxic activities

## Abstract

Natural daphnane diterpenoids, mainly distributed in plants of the *Thymelaeaceae* and *Euphorbiaceae* families, usually include a 5/7/6-tricyclic ring system with poly-hydroxyl groups located at C-3, C-4, C-5, C-9, C-13, C-14, or C-20, while some special types have a characteristic orthoester motif triaxially connectedat C-9, C-13, and C-14. The daphnane-type diterpenoids can be classified into five types: 6-epoxy daphnane diterpenoids, resiniferonoids, genkwanines, 1-alkyldaphnanes and rediocides, based on the oxygen-containing functions at rings B and C, as well as the substitution pattern of ring A. Up to now, nearly 200 daphnane-type diterpenoids have been isolated and elucidated from the *Thymelaeaceae* and *Euphorbiaceae* families. In-vitro and in-vivo experiments of these compounds have shown that they possess a wide range of biological activities, including anti-HIV, anti-cancer, anti-leukemic, neurotrophic, pesticidal and cytotoxic effects. A comprehensive account of the structural diversity is given in this review, along with the cytotoxic activities of daphnane-type diterpenoids, up to April 2019.

## 1. Introduction

Since the first daphnane diterpenoid characterized by a macrolactone motif was isolated from *Trigonostemon* reidioides [1], the daphnane diterpenoids have attracted the interest of many researchers because of their significant bioactive activities. Until now, nearly 200 natural products of daphnane-type diterpenoids have been isolated and identified, and they have shown good biological activities, including anti-HIV, anti-cancer, anti-leukemia, anti-hyperglycemic [2], neurotropic [3], insecticidal and cytotoxic [4] effects. Due to their rich pharmacological activities, especially strong anti-HIV activity and small cytotoxicity, daphnane-type diterpenoids have been employed in a range of clinical applications for a variety of clinical uses [5,6]. Studies have found that the natural daphnane-type diterpenoids usually embrace a 5/7/6-tricyclic ring system with poly-hydroxyl groups located at C-3, C-4, C-5, C-9, C-13, C-14, or C-20, while a special group also have a characteristic orthoester motif connected to C-9, C-13, and C-14. The daphnane-type diterpenoids can be categorized into five types (Figure 1): 6-epoxy daphnane diterpenoids, resiniferonoids, genkwanines, 1-alkyldaphnanes and rediocides, based on the substitution pattern of ring A and the oxygen-containing functions at rings B and C. Besides, 6-epoxy daphnane diterpenoids usually have a C-6α epoxy structure in ring B; resiniferonoids usually have an α-β unsaturated ketone structure in ring A; genkwanines usually have an α-β saturated ketone structure in ring A, but without a C-6α epoxy structure in ring B; 1-alkyldaphnanes usually have a saturated ring A, and a large ring between the end of the orthoester alkyl chain and C-1 of ring A; and rediocides usually have a 12-carbon macrolide structure between C-3 and C-16, and have a special C-9, C-12, and C-14 orthoester structure. The variety of daphnane-type diterpenoid structures have continued to widen with the discovery of unusual variations with the well-established skeleton. Owing to the unique skeleton and remarkable bioactive activities, daphnane-type diterpenoids have attracted many synthetic endeavors to construct a core structure. However, few papers have reported on the total synthesis of daphnane diterpenoids—isolation from natural plants is still the only source of obtaining daphnane diterpenoids. Considering the extensive interest in daphnane-type diterpenoids, we reviewed the structural and bioactive activities of daphnane-type diterpenoids, with an emphasis on the recent progress in structure identification and bioactive evaluation.

## 2. Occurrence

Natural daphnane-type diterpenoids are mainly distributed in species belonging to the *Thymelaeaceae* or *Euphorbiaceae* families (Table 1). These plants grow mainly in tropical and subtropical regions of Asia [7]. Previous chemical investigations on such species have led to the isolation of a number of structurally diverse diterpenoids [8]. Various daphnane-type diterpenoids have been isolated from some parts of the following plants: The twigs and leaves of *Trigonostemonthyrsoideum*, the roots of *Trigonostemonreidioides*, the stems of *Trigonostemon lii*, the twigs and leaves of *Trigonostemonchinensis* Merr, the stem barks of *Daphne giraldii*, the air-dried roots of *Euphorbia fischeriana*, the stems of *D. acutiloba*, the roots of *Lasiosiphonkraussianus*, the flower buds of *Daphne genkwa*, and the roots of *Maprouneaafricana* Muell. Arg., *Trigonostemonxyphophylloides*, *Wikstroemiaretusa*, *Trigonostemonhowii*, and *Stellerachamaejasme* L., and so on [9].

## 3. Species of Daphnane-Type Diterpenoids and Their Bioactive Activities

### 3.1. 6-Epoxy DaphnaneDiterpenoids

6-epoxy daphnane diterpenoids featurea C-6α epoxy structure in ring B and, occasionally, an α-β unsaturated ketone structure in ring A. In most cases, there is also a C-5β hydroxyl group and a C-20 hydroxyl group in ring B (Figure 2, Table 2). Compounds acutilobins A–G (**1**–**5**, **65**, **66**), wikstroemia factor M_1_ (**74**), genkwanineVIII (**69**), gniditrin (**14**), gnididin (**15**), gnidicin (**13**), daphnetoxin (**6**), yuanhuajine (**50**), kirkinine (**24**), excoecaria factor O_1_ (**8**), excoecaria toxin (**7**), and 14′-ethyltetrahydrohuratoxin(**51**) have been obtained from the stems of *D. acutiloba*. Acutilobins A–G have been shown to exhibit significant anti-HIV-1 activities, with EC_50_ below 1.5 μM [10]. Trigoxyphins A (**32**), B (**59**), and trigothysoid M (**63**) have been isolated from the twigs and leaves of *Trigonostemonthyrsoideum*. These compounds have been evaluated for anti-HIV activity by an assay of the inhibition of the cytopathic effects of HIV-1 and cytotoxicity against C8166 cells. However, only trigoxyphin A expressed weak anti-HIV-1 activity [11]. Compounds huratoxin (**20**) and wikstroelides A–D (**37**–**40**), H–J (**41**–**42, 56**), and L–N (**43**, **57**–**58**) have been obtained from the fresh bark of *Wikstroemiaretusa*. The orthoester compounds wikstroelides D and H, with palmitic acid at their 20-hydroxyl site, have shown the weakest cytotoxic activity [12]. Antitumor compounds genkwanin I (**64**) and orthobenzoate 2 (**70**) have been isolated from the flower buds of *Daphne genkwa*. Genkwanin I has been shown to be a potent cell growth inhibitor constituent [13]. Active ingredients genkwadane D (**9**), yuanhuadine (**47**), yuanhuafine (**45**), yuanhuacine (**49**), yuanhuahine (**44**), yuanhuapine (**61**), genkwadaphnine (**10**), isoyuanhuadine (**23**), and genkwanine M (**67**) were obtained from the flower buds of *Daphne genkwa*. Among them, yuanhuadine, genkwadaphnine, yuanhuafine, yuanhuapine, and genkwanine M have exhibited the strongest cytotoxic activities against the HT-1080 cell line (IC_50_ < 0.1 µM) [14]. Maprouneacin (**76**) has been isolated from the roots of *Maprouneaafricana* Muell. Arg, and has shown potent glucose-lowering properties when administered via the oral route. [15]. The compound trigonostempene C (**71**) has been obtained from the twigs and leaves of *Trigonostemonthyrsoideum*, but did not show any significant activity [16]. Compounds yuanhualine (**46**) and yuanhuagine (**48**) have been isolated from *Daphne genkwa*. In the analysis of signal transduction molecules, yuanhualine and yuanhuagine appear to suppress the activation of Akt, STAT3 and Src in human lung cancer cells, and also exert potent antiproliferative activity against anticancer-drug resistant cancer cells [17]. Gnidilatidin (**17**), gnidilatidin-20-palmitate (**18**), 1, 2α-dihydrodaphnetoxin (**62**), genkwadaphnin-20-palmitate (**11**) and gnidicin-20-palmitate (**19**) have successfully been obtained from the stems of *D. oleoidesSchreber* ssp. oleoides [18]. Trigoxyphins J and K (**33**–**34**) have been isolated from the stems of *Trigonostemonxyphophylloides*, and subsequently shown to be inactive against three tumor cell lines, specifically thehuman chronic myelogenous leukemia cell line (K562), the human gastric carcinoma cell line (SGC-7901), and human hepatocellular carcinoma (BEL-7402) (IC_50_ value > 10 μM) [19]. Genkwanine N (**68**) has been obtained from the dried flower buds of *Daphne genkwa*, and the compound with esterification of the 20-hydroxyl has shown weak toxicity [20]. Trigonosin B (**73**) has beenisolated from the roots of *Trigonostemonthyrsoideum* [21], whilecompounds hirseins A and B (**21**–**22**) have been isolated from *Thymelaeahirsuta*. Hirseins A and B have shown inhibition of melanogenesis in B16 murine melanoma cells [22]. Glabrescin(**12**) and Montanin (**26**) have been obtained from *Neoboutoniaglabrescens* [23]. Kirkinine D (**25**) and synaptolepisfactor K_7_ (**28**) have been isolated from the *S.kirkii* [24]. Wikstrotoxin C (**35**) has been isolated from *W.monticola*. The compound 2α-dihydro-20-palimoyldaphnetoxin (**52**) has been isolated from the *D.tangutica*, while gnidiglaucin (**16**) has been obtained from *P.elongata* [24]. Trigoxyphin C (**60**) has been obtained from *T.xyphophylloides*, and tested against BEL-7402 cells (human hepatocellular carcinoma), where in it has been shown to be inactive (IC_50_ value > 10 µM was defined as inactive) [25]. Trigonosin A (**72**) has been isolated from *T.thyrsoideum*, and shown to exhibitin significant inhibitory activity against specific tumor cells (IC_50_ >10 μM) [21]. Isovesiculosin and vesiculosin (**54**–**55**) have been isolated from *D.vesiculosum* [26]. Genkwanine O (**75**) has been obtained from *D.genkwa*. Compound daphnegiraldigin (**53**) has been isolated from the stem barks of *Daphne giraldii* [27]. Simplexin(**27**) has been obtained from *Stellerachamaejasme* L. [5]. Compounds trigochinins G–I (**29**–**31**) have been isolated from the twigs and leaves of *Trigonostemonchinensis* Merr [28].

### 3.2. Resiniferonoids

Relative to 6-epoxy daphnane diterpenoids, there is no C-6α epoxy structure in ring B forresiniferonoids. However, resiniferonoids do possess an α-β unsaturated ketone structure in ring A (Figure 3, Table 3). Compounds 4β, 9α, 20- trihydroxy- 13, 15- secotiglia- 1,6- diene- 3,13- dione 20-*O*-β-d- [6-galloyl] glu- copyranoside (**86**) and euphopiloside A (**84**) have beenisolated from the air-dried roots of *Euphorbia fischeriana*, and display moderate inhibitory effects against α-glucosidase in in-vitro bioassays [29]. Yuanhuatine (**78**) has been isolated from the flower buds of *Daphne genkwa* [14]. Compounds daphneresiniferins A and B (**80**–**81**) have been obtained from the flower buds of *Daphne genkwa*. A study found that daphneresiniferin A was able to dependently inhibit melanin production [30]. Genkwanine L (**77**) has been isolated from the bud of *Daphne genkwa* [31]. Euphopiloside B (**83**), langduin A (**85**) and phorbol (**87**) have been obtained from the *Euphorbia Pilosa* [32], while compounds genkwadane A (**79**) and yuanhuaoate B (**82**) have been isolated from the flower buds of *Daphne genkwa* [14].

### 3.3. Genkwanines

Relative to 6-epoxy daphnane diterpenoids and resiniferonoids, genkwanines have an α-β saturated ketone structure in ring A, but do not possess a C-6α epoxy structure in ring B (Figure 4, Table 4). Compound trigoxyphin H (**100**) has been isolated from the twigs of *Trigonostemonxyphophylloides* [33]. The active ingredients trigothysoids A–L (**122**–**124**, **96**–**99**, **139**–**141**, **131**,**128**), trigochinins A–E (**145**–**146**, **130**, **147**–**148**), andtrigonothyrins D, E (**143**–**144**) and G (**121**) have been obtained from the twigs and leaves of *Trigonostemonthyrsoideum*. These compounds have been evaluated for their anti-HIV activity usingan assay to determine their inhibition of the cytopathic effects of HIV-1 and their cytotoxicity against C8166 cells. Amongst them, trigothysoid A and L exhibited moderate anti-HIV-1 activity; andtrigothysoid C and K andtrigochinins A, B and D expressed weak anti-HIV-1 activity [11]. Trigolins A–G (**132**–**138**) and trigonothyrin F (**107**) have been isolated from the stems of *Trigonostemon lii*. Trigolins A, G, H, and K have been shown to exhibit modest anti-HIV-1 activity with EC_50_ values of 2.04, 9.17, 11.42, and 9.05l µg/mL, respectively [34]. Compound trigochinin F (**149**) has been obtained from the twigs and leaves of *Trigonostemonchinensis* Merr, and has shown strong inhibition of HL-60 tumor cell lines [28]. Trigonothyrins A–C (**125**–**127**) have been isolated from the stems of *Trigonostemonthyrsoideum* [6]. Among them, trigonothyrin C has shown significant activity to prevent the cytopathic effects of HIV-1 in C8166 cells, with an EC_50_ value of 2.19 µg/mL [35]. Compounds genkwanines F, I, and J (**93**, **113**, **114**) have been isolated from the flower buds of *Daphne genkwa* [14]. Genkwanine H **(95)** has been obtained from the flower buds of *Daphne genkwa*, and the compound has been shown to dependently inhibit melanin production [30]. Compounds trigonostempenes A (**150**) and B (**129**) have been isolated from the twigs and leaves of *Trigonostemonthyrsoideum*. Studies have shown that the discovery of these NO inhibitory daphnane diterpenoids—including compound trigonostempene A—which possess IC_50_ values comparable topositive controls may have the potential to be developed as anti-neuroinflammatory agents for alzheimer disease (AD) and other related neurological disorders [16]. Most inhibitors of acetylcholinesterase (AchE) are alkaloids that often possess several side effects, whereas these daphnane-type diterpenoids do not belong to the class of alkaloids, and therefore they may constitute novel active AChE inhibitors with fewer side effects. It is important to search for new AChE inhibitors not belonging to this structural class [36,37]. Genkwanines A–E (**88**–**92**), G (**94**), I (**113**), and K (**115**) have been obtained from the bud of *Daphne genkwa*. Among these compounds, genkwanine D has been shown to exhibit strong activity to inhibit the endothelium cell HMEC at IC_50_ levels of 2.90–15.0 μM [31]. Compounds trigoxyphins U and W (**105**–**116**) have been isolated from the twigs of *Trigonostemonxyphophylloides*. Trigoxyphin W has shown modest cytotoxicity against BEL-7402, SPCA-1 and SGC-7901, with IC_50_ values of 5.62, 16.79 and 17.19 µM, respectively [33]. Trigonosins C–D (**106**, **142**) have been obtained from the roots of *Trigonostemonthyrsoideum* [21]. Trigoxyphin I **(104)** has been isolated from the *Trigonostemonxyphophylloides* [38]. Compounds trigohownins D and E (**101**–**102**), and trigohownins A–C (**108**–**110**) and F–I (**117**–**120**) have been obtained from the *Trigonostemonhowii*. Among them, trigohownins A and D have been shown to exhibit moderate cytotoxic activity against the HL-60 tumor cellline, with IC_50_ values of 17.0 and 9.3 μM, respectively [39]. Trigoxyphins D–F (**111**–**112**, **103**) have been isolated from *Trigonostemonxyphophylloides*, with all three compounds found to be inactive against BEL-7402 cells (IC_50_ value > 10 µM) [25].

### 3.4. 1-Alkyldaphnanes

1-alkyldaphnanes have a large ring between the end of the orthoester alkyl chain and C-1 of ring A (Figure 5, Table 5). Pimelea factors S_6_ (**168**) and S_7_ (**169**) have been isolated from the flower buds of *Wikstroemiachamaedaphne* and have shown moderate cytotoxic activities against human myeloid leukemia HL-60, hepatocellular carcinoma SMMC-7721, lung cancer A549, breast cancer MCF-7, and colon cancer SW480 [1]. Compound pimelea factor P_2_ (**155**) has been obtained from the fresh bark of *Wikstroemiaretusa*, and has been shown to exhibit cytotoxicity in 10 cell lines (including HeLa, HepG2, HT-1080, HCT116, A375-S2, MCF-7, A549, U-937, K562 and HL60 cell lines) [14]. Wikstroelides E–G, K and O (**163**–**167**) have been isolated from the fresh bark of *Wikstroemiaretusa*. Among them, compound wikstroelide E has been shown to exhibit the highest activity against cell lines PC-6 (human lung cancer cell line) and P388 (mouse leukaemia cell line), followed by wikstroelides A and J, which have the orthoester group without a fatty acid at the 20-hydroxyl [12]. Compounds stelleralides A–C (**151**–**152**, **174**) and gnidimacrin (**153**) have been isolated from the *Stellerachamaejasme* L. [5]. Genkwadane B (**154**), pimelotides A and C (**170**, **172**), and genkwadane C (**156**) have been isolated from the flower buds of *Daphne genkwa* [14]. Compounds wikstroelides R–T (**157**–**159**) have been obtained from the flower buds of *Wikstroemiachamaedaphne*. Wikstroelide R has been shown to have moderate cytotoxic activities against human cancer cell lines [1]. Compounds kirkinines B, C, and E (**160**–**162**) were isolated from *Synaptolepiskirkii*. Pimelotides B and D (**171**, **173**) have beenobtained from *Pelongata* [40].

### 3.5. Rediocides

Rediocides usually have a 12-carbon macrolide structure between C-3 and C-16, and have a special C-9, C-12, and C-14 orthoester structure (Figure 6, Table 6). The active compounds trigothysoids N–P (**182**–**184**), rediocides A, C, and F (**176**–**177**, **179**), and trigonosin F (**181**) have been obtained from the twigs and leaves of *Trigonostemonthyrsoideum*. Amongst them, compounds trigothysoid N, rediocides A, C, and F, and trigonosins F have shownpotent anti-HIV-1 activity, with EC_50_ values ranging from 0.001 to 0.015 nM. Additionally, trigothysoid O has been shown to exhibit moderate anti-HIV-1 activity [11], while rediocide A has shown potent activities against mosquito larvae in an in-vitro assay study and against fleas (Ctenocephalides felis) in an artificial membrane feeding system, exhibiting LD_90_ values of 1 and 0.25 ppm, respectively [39]. Trigochilides A and B (**175**, **186**) have been isolated from the twigs and leaves of *Trigonostemonchinensis* Merr. Trigochilide A has shown modest cytotoxicity against HL-60 (human leukemia) and BEL-7402 (human hepatoma), with demonstrated IC_50_ values of 3.68 and 8.22 µM, respectively, whereas compound trigochilide B has only been shown to exhibit weak cytotoxicity against two tumor cell lines, with IC_50_ values of 33.35 and 54.85 µM [1]. Compound rediocide E (**178**) has been obtained from the roots of *Trigonostemonreidioides*, and has shown significant acaricidal activity on D. pteronyssinus [40]. Trigonosin E (**180**) and trigonostempene D (**185**) have beenisolated from the twigs and leaves of *Trigonostemonthyrsoideum* [16,21]. Rediocides B, G, and D (**187**–**189**) have been isolated from the *Trigonostemonreidioides*, and have been evaluated for their insecticidal properties in an anti-flea artificial membrane feeding assay (as detailed earlier). In this assay, rediocides B and D exhibited LD_90_ values of 0.25 and 0.5 ppm, respectively, and thus were equipotent with rediocide A (LD_90_ 0.25 ppm) [41].

## 4. Conclusions

It can be concluded that the bioactive activities of daphnane-type diterpenoids is obviously related to structure types. The most important points of them are the following: (1) The orthoester groups at C-9, C-13 and C-14 are essential to the cytotoxic activity. Daphnane-type diterpenoids with orthoester groups at C-9, C-13, and C-14 usually have stronger activity than daphnane-type diterpenoids with orthoester groups at C-9, C-12, and C-14 or C-12, C-13 and C-14. The absence of the orthoester group is unhelpful to the cytotoxic activity. (2) Specific to the 6-epoxyl groups, free 20-hydroxyl and 3-carbonyl are important for their activities. (3) Side chains at C-10 are crucial for cytotoxic activities. Generally speaking, long C-10 alkyl chains are more important than phenyl at C-10. Interestingly, the structure with macro-lactones exhibited much stronger activity than the others. Due to the rich activities of daphnane-type diterpenoids, researchers have not stopped exploring and researching such compounds and their bioactive activities from plants.

## Figures and Tables

**Figure 1 molecules-24-01842-f001:**
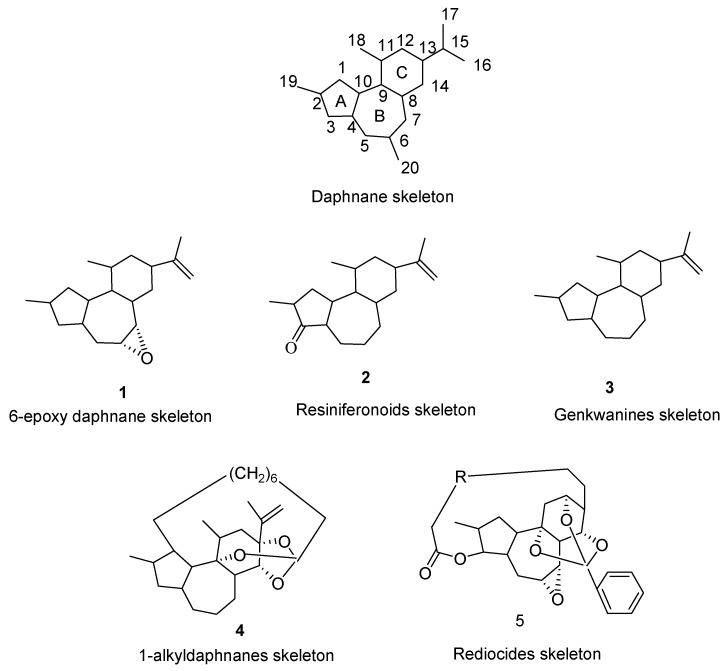
The kinds of daphnane-type diterpenoids skeleton.

**Figure 2 molecules-24-01842-f002:**
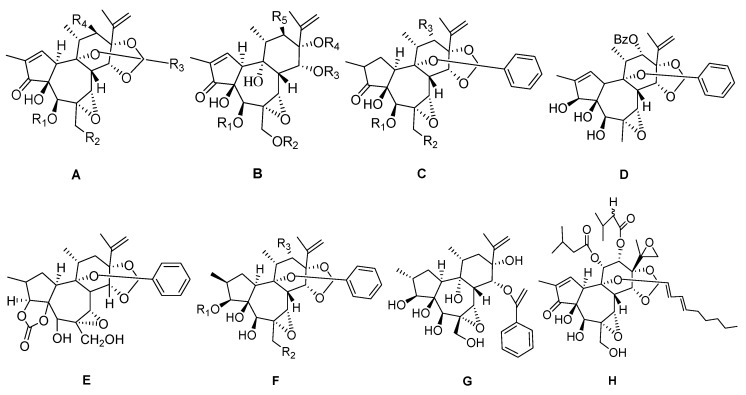
Eight types (**A**–**H**) of 6-epoxy daphnane skeletons.

**Figure 3 molecules-24-01842-f003:**
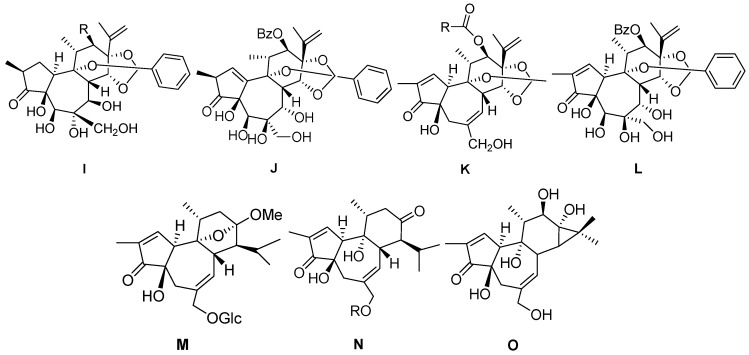
Seven types (**I**–**O**) of resiniferonoids skeletons.

**Figure 4 molecules-24-01842-f004:**
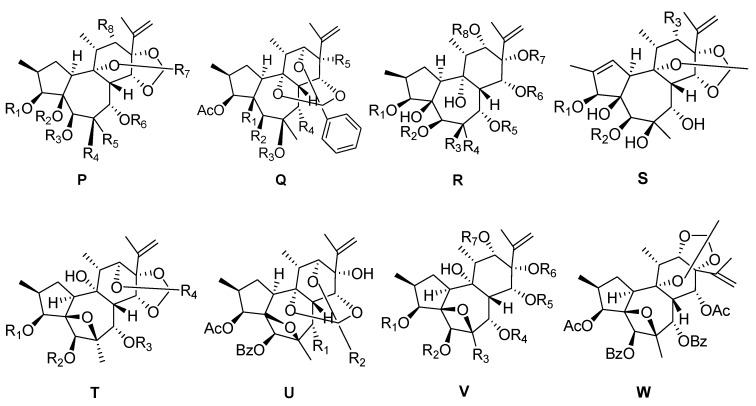
Eight types (**P**–**W**) of genkwanines skeletons.

**Figure 5 molecules-24-01842-f005:**
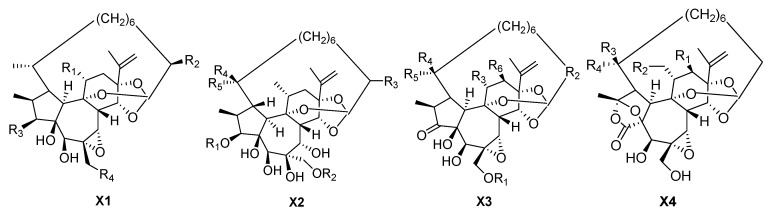
Four types (**X1**–**X4**) of 1-alkyldaphnanes skeletons.

**Figure 6 molecules-24-01842-f006:**
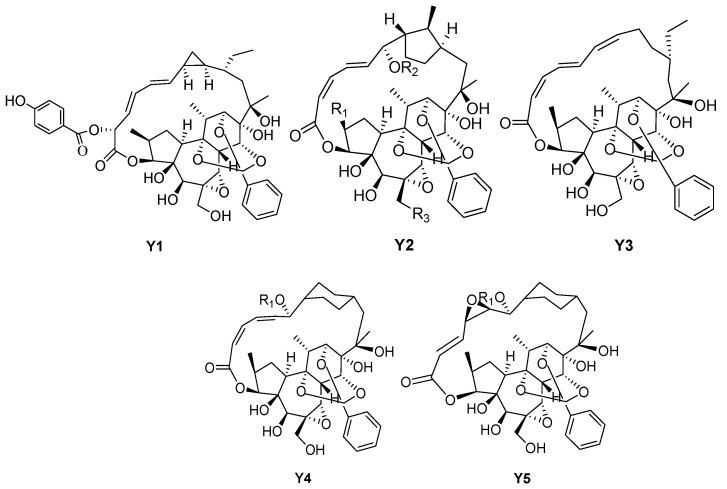
Five types (**Y1**–**Y5**) of rediocides skeletons.

**Table 1 molecules-24-01842-t001:** The species of daphnane-type diterpenoids.

Types of Diterpenoids	Species	Medication Site
6-epoxy daphnane diterpenoids	*D. acutiloba*	Usually their effective part is roots, stems, twigs and leaves, flower buds, fresh bark.
	*Trigonostemonthyrsoideum*	
	*Wikstroemiaretusa*	
	*Daphne genkwa*	
	*D. oleoidesSchreber* ssp. oleoides	
	*Trigonostemonxyphophylloides*	
	*Thymelaeahirsuta*	
	*Neoboutoniaglabrescens*	
	*S.kirkii*	
	*W.monticola*	
	*D.tangutica*	
	*P.elongata*	
	*T.xyphophylloides*	
	*T.thyrsoideum*	
	*D.vesiculosum*	
	*Stellerachamaejasme* L.	
	*Trigonostemonchinensis* Merr	
Resiniferonoids	*Euphorbia fischeriana*	Generally, the roots and flower budsistheir effective part.
	*Daphne genkwa*	
	*Euphorbia pilosa*	
Genkwanines	*Trigonostemonxyphophylloides*	Usually their effective part isroots, stems, twigs and leaves, flower buds.
	*Trigonostemonthyrsoideum*	
	*Trigonostemon lii*	
	*Trigonostemonchinensis* Merr	
	*Daphne genkwa*	
	*Trigonostemonhowii*	
1-alkyldaphnanes	*Wikstroemiachamaedaphne*	Usually, the flower buds and fresh bark is their effective part.
	*Wikstroemiaretusa*	
	*Stellerachamaejasme* L.	
	*Daphne genkwa*	
	*Synaptolepiskirkii*	
	*P.elongata*	
Rediocides	*Trigonostemonthyrsoideum*	Generally, their effective part is roots, twigs and leaves.
	*Trigonostemonchinensis* Merr	
	*Trigonostemonreidioides*	

**Table 2 molecules-24-01842-t002:** Reported structures of 6-epoxy daphnane skeletons.

No.	Name	R_1_	R_2_	R_3_	R_4_	R_5_	Type
1	Acutilobin A	H	OH	Ph	OCO(CH=CH)_2_COC(CH_2_)_2_CH_3_	–	A
2	Acutilobin B	H	OH	Ph	OCO(CH=CH)_3_CHCH_2_CH_3_OH	–	A
3	Acutilobin C	H	OH	(CH=CH)_3_(CH_2_)_2_CH_3_	OCOCH=CHPhCH_3_OH	–	A
4	Acutilobin D	H	OH	(CH=CH)_2_(CH_2_)_4_CH_3_	OCOCH=CHPhCH_3_OH	–	A
5	Acutilobin E	H	OH	Ph	OCOCH=CHPhCH_3_OH	–	A
6	Daphnetoxin	H	OH	Ph	H	–	A
7	Excoecaria toxin	H	OH	(CH=CH)_2_(CH_2_)_4_CH_3_	H	–	A
8	Excoecaria factor O_1_	H	OH	(CH=CH)_3_(CH_2_)_2_CH_3_	H	–	A
9	Genkwadane D	H	OH	(CH=CH)_2_(CH_2_)_4_CH_3_	OCOCH(CH_3_)_2_	–	A
10	Genkwadaphnine	H	OH	Ph	OBz	–	A
11	Genkwadaphnin-20-palmitate	H	OCO(CH_2_)_14_CH_3_	Ph	OCOPh	–	A
12	Glabrescin	H	OCOCH_2_(CH_2_)_13_CH_3_	(CH_2_)_10_CH_3_	H	–	A
13	Gnidicin	H	OH	Ph	OCOCH=CHPh	–	A
14	Gniditrin	H	OH	Ph	OCO(CH=CH)_3_(CH_2_)_2_CH_3_	–	A
15	Gnididin	H	OH	Ph	OCO(CH=CH)_2_(CH_2_)_4_CH_3_	–	A
16	Gnidiglaucin	H	OH	(CH_2_)_8_CH_3_	OAc	–	A
17	Gnidilatidin	H	OH	(CH=CH)_2_(CH_2_)_4_CH_3_	OCOPh	–	A
18	Gnidilatidin-20-palmitate	H	OCO(CH_2_)_14_CH_3_	(CH=CH)_2_(CH_2_)_4_CH_3_	OCOPh	–	A
19	Gnidicin-20-palmitate	H	OCO(CH_2_)_14_CH_3_	Ph	OCOCH=CHPh	–	A
20	Huratoxin	H	OH	(CH=CH)_2_(CH_2_)_8_CH_3_	H	–	A
21	Hirsein A	H	OH	CH=CH(CH_2_)_4_CH_3_	OCOCH=CHPh	–	A
22	Hirsein B	H	OH	CH=CH(CH_2_)_4_CH_3_	OCOCH=CHPhOH	–	A
23	Isoyuanhuadine	H	OH	(CH=CH)_2_(CH_2_)_4_CH_3_	OAc	–	A
24	Kirkinine	H	OH	CH=CH(CH_2_)_12_CH_3_	OAc	–	A
25	Kirkinine D	H	OH	(CH=CH)_3_(CH_2_)_2_CH_3_	OAc	–	A
26	Montanin	H	OH	(CH_2_)_10_CH_3_	H	–	A
27	Simplexin	H	OH	(CH_2_)_8_CH_3_	H	–	A
28	Synaptolepisfactor K_7_	H	OH	CH=CH(CH_2_)_12_CH_3_	H	–	A
29	Trigochinin G	H	H	Ph	OCOCH_2_CH(CH_3_)_2_	–	A
30	Trigochinin H	H	H	Ph	OCOC_6_H_4_(4-OH)	–	A
31	Trigochinin I	H	H	Ph	OCOC_6_H_3_(_3_-OMe)(4-OH)	–	A
32	Trigoxyphin A	H	H	Ph	OBz	–	A
33	Trigoxyphin J	H	OH	CH_3_	OCO(CH_2_)_14_CH_3_	–	A
34	Trigoxyphin K	H	H	Ph	OBz	–	A
35	Wikstrotoxin C	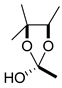	OH	(CH=CH)_2_(CH_2_)_4_CH_3_	OAc	–	A
36	Wikstrotoxin D	H	OH	n-C_9_H_19_	H	–	A
37	Wikstroelide A	H	OH	(CH=CH)_2_(CH_2_)_8_CH_3_	OAc	–	A
38	Wikstroelide B	H	OH	(CH=CH)_2_(CH_2_)_9_CH_3_	OAc	–	A
39	Wikstroelide C	H	O-trans-5-pentadecenoic acid	(CH=CH)_2_(CH_2_)_8_CH_3_	OAc	–	A
40	Wikstroelide D	H	O-palmitic acid	(CH=CH)_2_(CH_2_)_8_CH_3_	OAc	–	A
41	Wikstroelide H	H	OH	(CH=CH)_2_(CH_2_)_6_CH_3_	OAc	–	A
42	Wikstroelide I	H	O-palmitic acid	(CH=CH)_2_(CH_2_)_9_CH_3_	OAc	–	A
43	Wikstroelide L	H	OH	(CH=CH)_2_(CH_2_)_8_CH_3_	OAc	–	A
44	Yuanhuahine	H	OH	(CH=CH)_2_(CH_2_)_4_CH_3_	OCOCH_2_CH_3_	–	A
45	Yuanhuafine	H	H	Ph	OAc	–	A
46	Yuanhualine	H	OH	(CH=CH)_2_(CH_2_)_4_CH_3_	OCO(CH_2_)_2_CH_3_	–	A
47	Yuanhuadine	H	OH	(CH=CH)_2_(CH_2_)_4_CH_3_	OAc	–	A
48	Yuanhuagine	H	OH	(CH=CH)(CH_2_)_2_CH_3_	OCOCH_3_	–	A
49	Yuanhuacine	H	OH	(CH=CH)_2_(CH_2_)_4_CH_3_	OBz	–	A
50	Yuanhuajine	H	OH	(CH=CH)_3_(CH_2_)_2_CH_3_	OBz	–	A
51	14′-ethyltetrahydrohuratoxin	H	OH	(CH_2_)_14_CH_3_	H	–	A
52	2α-dihydro-20-palimoyldaphnetoxin	H	OH	CH=CH(CH_2_)_6_CH_3_	OAc	–	A
53	Daphnegiraldigin	H	OH	COPh	H	H	B
54	Isovesiculosin	Ac	Ac	Ac	CO(CH=CH)_2_(CH_2_)_4_CH_3_	H	B
55	Vesiculosin	H	H	CO(CH=CH)_2_(CH_2_)_4_CH_3_	H	H	B
56	Wikstroelide J	H	H	CO(CH=CH)_2_(CH_2_)_8_CH_3_	H	OAc	B
57	Wikstroelide M	H	H	CO(CH=CH)_2_(CH_2_)_8_CH_3_	H	H	B
58	Wikstroelide N	H	H	CO(CH=CH)_2_(CH_2_)_9_CH_3_	H	H	B
59	Trigoxyphin B	H	H	OBz	–	–	C
60	Trigoxyphin C	Ac	H	OBz	–	–	C
61	Yuanhuapine	H	OH	OAc	–	–	C
62	1,2α-dihydrodaphnetoxin	H	OH	H	–	–	C
63	Trigothysoid M	–	–	–	–	–	D
64	Genkwanin I	–	–	–	–	–	E
65	Acutilobin F	CO(CH=CH)_3_(CH_2_)_2_CH_3_	OH	H	–	–	F
66	Acutilobin G	COCH=CHPh	OH	H	–	–	F
67	Genkwanine M	H	OBz	H	–	–	F
68	Genkwanine N	Bz	OH	H	–	–	F
69	GenkwanineVIII	COPh	OH	H	–	–	F
70	Orthobenzoate 2	H	OH	H	–	–	F
71	Trigonostempene C	H	H	OH	–	–	F
72	Trigonosin A	H	H	OBz	–	–	F
73	Trigonosin B	H	OH	OBz	–	–	F
74	Wikstroemia factor M_1_	CO(CH=CH)_2_(CH_2_)_4_CH_3_	OH	H	–	–	F
75	Genkuanine O	–	–	–	–	–	G
76	Maprouneacin	–	–	–	–	–	H

**Table 3 molecules-24-01842-t003:** Reported structures of resiniferonoids skeletons.

No.	Name	R	Type
77	Genkwanine L	OAc	I
78	Yuanhuatine	OBz	I
79	Genkwadane A	–	J
80	Daphneresiniferin A	Me	K
81	Daphneresiniferin B	Ph	K
82	Yuanhuaoate B	–	L
83	Euphopiloside B	–	M
84	Euphopiloside A	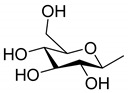	N
85	Langduin A	H	N
86	4β,9α,20-trihydroxy-13,15-secotiglia-1,6-diene-3,13-dione20-*O*-β-d-[6-galloyl]glu-copyranoside	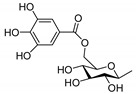	N
87	Phorbol	–	O

**Table 4 molecules-24-01842-t004:** Reported structures of genkwanines skeletons.

No.	Name	R_1_	R_2_	R_3_	R_4_	R_5_	R_6_	R_7_	R_8_	Type
88	Genkwanine A	H	H	H	OH	CH_2_OH	H	Ph	H	P
89	Genkwanine B	CO(CH=CH)_2_(CH_2_)_4_CH_3_	H	H	OH	CH_2_OH	H	Ph	H	P
90	Genkwanine C	CO(CH=CH)_3_(CH_2_)_2_CH_3_	H	H	OH	CH_2_OH	H	Ph	H	P
91	Genkwanine D	Bz	H	H	OH	CH_2_OH	H	Ph	H	P
92	Genkwanine E	H	H	H	OH	CH_2_OCO(CH=CH)_3_(CH_2_)_2_CH_3_	H	Ph	H	P
93	Genkwanine F	H	H	H	OH	CH_2_OCO(CH=CH)_2_(CH_2_)_4_CH_3_	H	Ph	H	P
94	Genkwanine G	H	H	H	OH	CH_2_COO(CH=CH) (CH_2_)_6_CH_3_	H	Ph	H	P
95	Genkwanine H	H	H	H	OH	CH_2_OBz	H	Ph	H	P
96	Trigothysoid D	H	H	H	OH	Me	H	Me	OBz	P
97	Trigothysoid E	Ac	H	H	OH	Me	H	Me	OBz	P
98	Trigothysoid F	H	H	Ac	OH	Me	H	Me	OBz	P
99	Trigothysoid G	Ac	H	Bz	OH	ME	H	Me	OBz	P
100	Trigoxyphin H	Ac	H	Ac	OCOPh	Me	Ac	Ph	OAc	P
101	Trigohownin D	Ac	Bz	Ac	OH	Me	Ac	Ph	OAc	P
102	Trigohownin E	Ac	H	Bz	OH	Me	Ac	Me	OBz	P
103	Trigoxyphin F	Ac	H	Ac	OBz	Me	Ac	Ph	OH	P
104	Trigoxyphin I	Ac	H	Ac	OCOPh	Me	Ac	Ph	Ac	P
105	Trigoxyphin U	Ac	H	Ac	Me	OCOPh	Ac	ME	OCOPh	P
106	Trigonosin C	H	H	H	OH	Me	H	Ph	OBz	P
107	Trigonothyrin F	H	H	H	OH	Me	H	Ph	H	P
108	Trigohownin A	OAc	OH	Bz	OAc	OH	–	–	–	Q
109	Trigohownin B	OBz	OAC	H	OAc	OH	–	–	–	Q
110	Trigohownin C	OH	OAC	Bz	OH	OH	–	–	–	Q
111	Trigoxyphin D	OH	OAC	Bz	OAc	OH	–	–	–	Q
112	Trigoxyphin E	H	OAC	Bz	OAc	OAc	–	–	–	Q
113	Genkwanine I	H	H	OH	CH_2_OH	H	Bz	H	H	R
114	Genkwanine J	H	H	OH	CH_2_OCO(CH=CH)_2_(CH_2_)_4_CH_3_	H	Bz	H	H	R
115	Genkwanine K	H	H	OH	CH_2_Bz	H	Bz	H	H	R
116	Trigoxyphin W	Ac	Ac	Me	OCOPh	H	H	COPh	H	R
117	Trigohownin F	Ac	Ac	OBz	Me	Ac	H	Bz	OH	R
118	Trigohownin G	Ac	Ac	OBz	Me	Ac	Ac	Bz	OH	R
119	Trigohownin H	Ac	Ac	OBz	Me	Ac	Bz	Ac	OH	R
120	Trigohownin I	Ac	Bz	OH	Me	Ac	Ac	Bz	OH	R
121	Trigonothyrin G	Ac	H	OCOPh	–	–	–	–	–	S
122	Trigothysoid A	H	H	OBz	–	–	–	–	–	S
123	Trigothysoid B	Ac	Bz	OBz	–	–	–	–	–	S
124	Trigothysoid C	H	Ac	OBz	–	–	–	–	–	S
125	Trigonothyrin A	Bz	Ac	Bz	Me	–	–	–	–	T
126	Trigonothyrin B	H	Bz	Bz	Me	–	–	–	–	T
127	Trigonothyrin C	Ac	Bz	Bz	Me	–	–	–	–	T
128	Trigothysoid L	Ac	Bz	Ac	Ph	–	–	–	–	T
129	Trigonostempene B	Ac	Ac	Bz	Me	–	–	–	–	T
130	Trigochinin C	OAc	Ph	–	–	–	–	–	–	U
131	Trigothysoid K	OBz	Me	–	–	–	–	–	–	U
132	Trigolins A	H	Bz	Me	Ac	Ac	H	Bz	–	V
133	Trigolins B	Ac	Bz	Me	Ac	H	H	Bz	–	V
134	Trigolins C	Ac	Bz	Me	Ac	Bz	H	H	–	V
135	Trigolins D	Ac	Bz	Me	Ac	Ac	H	Bz	–	V
136	Trigolins E	Ac	Bz	Me	Bz	Ac	H	Ac	–	V
137	Trigolins F	Ac	Ac	Me	Bz	Ac	H	Bz	–	V
138	Trigolins G	H	Bz	Me	Bz	AC	H	Bz	–	V
139	Trigothysoid H	Ac	Ac	CH_2_OAc	Ac	Ac	Bz	Ac	–	V
140	Trigothysoid I	Ac	Ac	CH_2_OAc	Ac	Ac	H	Bz	–	V
141	Trigothysoid J	Ac	Bz	Me	Ac	Ac	H	Bz	–	V
142	Trigonosin D	H	H	Me	Ac	Ac	COPh	Ac	–	V
143	Trigonothyrin D	Ac	Ac	Me	Ac	Ac	COPh	Ac	–	V
144	Trigonothyrin E	H	Ac	Me	Ac	Ac	COPh	Ac	–	V
145	Trigochinin A	H	Bz	Me	Ac	Ac	COPh	Ac	–	V
146	Trigochinin B	Ac	Bz	Me	Ac	Ac	COPh	Ac	–	V
147	Trigochinin D	H	Bz	Me	Ac	Ac	Bz	Ac	–	V
148	Trigochinin E	Ac	Bz	Me	Ac	Ac	Bz	Ac	–	V
149	Trigochinin F	Ac	Ac	Ac	Ac	Ac	Bz	Ac	–	V
150	Trigonostempene A	–	–	–	–	–	–	–	–	W

**Table 5 molecules-24-01842-t005:** Reported structures of 1-alkyldaphnanes skeletons.

No.	Name	R_1_	R_2_	R_3_	R_4_	R_5_	R_6_	Type
151	Stelleralide A	CH_2_OAc	OH	OBz	OH	–	–	X1
152	Stelleralide B	CH_2_OBz	H	OBz	OH	–	–	X1
153	Gnidimacrin	CH_2_OBz	OH	OBz	OH	–	–	X1
154	Genkwadane B	Me	H	OH	OBz	–	–	X1
155	Pimelea factor P_2_	CH_2_OH	H	OBz	OH	–	–	X1
156	Genkwadane C	H	benzoyl	H	H	Me	–	X2
157	Wikstroelide R	H	benzoyl	OH	H	Me	–	X2
158	Wikstroelide S	benzoyl	H	H	Me	H	–	X2
159	Wikstroelide T	H	trans-cinnamoyl	H	H	Me	–	X2
160	Kirkinine B	H	CH=CH(CH_2_)_5_	Me	H	Me	H	X3
161	Kirkinine C	H	CH=CH(CH_2_)_5_	Me	H	Me	OAc	X3
162	Kirkinine E	H	CH=CH(CH_2_)_5_	Me	OH	Me	H	X3
163	Wikstroelide E	H	CH_2_	Me	H	Me	H	X3
164	Wikstroelide F	H	CH_2_	CH_2_OBz	H	Me	H	X3
165	Wikstroelide G	palmitic acid	CH_2_	CH_2_OBz	H	Me	H	X3
166	Wikstroelide K	CO(CH_2_)14CH_3_	CH_2_	CH_2_OBz	Me	H	H	X3
167	Wikstroelide O	H	CH_2_	CH_2_OBz	Me	H	H	X3
168	Pimelea factor S_6_	OH	CH_2_	Me	H	Me	H	X3
169	Pimelea factor S_7_	OH	CH_2_	Me	Me	H	H	X3
170	Pimelotide A	H	H	Me	H	–	–	X4
171	Pimelotide B	OAc	H	H	Me	–	–	X4
172	Pimelotide C	H	H	H	Me	–	–	X4
173	Pimelotide D	OAc	H	Me	H	–	–	X4
174	Stelleralide C	H	OBz	Me	H	–	–	X4

**Table 6 molecules-24-01842-t006:** Reported structures of rediocides skeletons.

No.	Name	R_1_	R_2_	R_3_	Type
175	Trigochilide A	–	–	–	Y1
176	Rediocide A	Me	COCH_2_CH(CH_3_)_2_	OH	Y2
177	Rediocide C	Me	Bz	OH	Y2
178	Rediocide E	H	COCH_2_CH(CH_3_)_2_	OH	Y2
179	Rediocide F	H	Bz	OH	Y2
180	Trigonosin E	Me	COPh	OH	Y2
181	Trigonosin F	Me	COPh	OH	Y2
182	Trigothysoid N	Me	COCH_2_CH(CH_3_)_2_	OH	Y2
183	Trigothysoid O	Me	COPh	H	Y2
184	Trigothysoid P	Me	COCH_2_CH(CH_3_)_2_	H	Y2
185	Trigonostempene D	Me	Val	H	Y2
186	Trigochilide B	–	–	–	Y3
187	Rediocide B	COCH_2_CH(CH_3_)_2_	–	–	Y4
188	Rediocide G	Bz	–	–	Y4
189	Rediocide D	COCH_2_CH(CH_3_)_2_	–	–	Y5

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
