# Peer review of "A Review on Daphnane-Type Diterpenoids and Their Bioactive Studies"

_molecules, 2019, doi:10.3390/molecules24091842_

Round 1

Reviewer 1 Report

To be frank, lovely and blinded. In other words, I really liked Your review article. Therefore, my recommendation is "Accept after minor revision".

If possible, please kindly improve a bit the English language.

Also, You may kindly consider citing of the following references due to highlighting the importance of anti-AChE activity screening versus daphnane-type diterpenoids in the time to come (as part of the further perspectives in this particular research field):

J Nat Prod. 2008 Nov;71(11):1850-3. doi: 10.1021/np800318m.

Eur J Med Chem. 2016 Oct 21;122:326-338. doi: 10.1016/j.ejmech.2016.06.036.

Once again, my sincere congrats to all authors. Without a doubt, Your promising article will be quite well recognised within natural product research communities worldwide due to its straightforward communication.

Last but not least, very best of (research) luck ahead!

Author Response

Response to Reviewer 1 Comments

Point 1: If possible, please kindly improve a bit the English language.

Response 1: It is really correct as reviewer suggested that improve the English language. We tried our best to improve the quality of English sentences.

Point 1: You may kindly consider citing of the following references due to highlighting the importance of anti-AChE activity screening versus daphnane-type diterpenoids in the time to come (as part of the further perspectives in this particular research field):

- J Nat Prod. 2008 Nov;71(11):1850-3. doi: 10.1021/np800318m.

- Eur J Med Chem. 2016 Oct 21;122:326-338. doi: 10.1016/j.ejmech.2016.06.036.

Response 2: Your suggestion is very meaningful. I have carefully read the two references and have added relevant anti-AChE activity to the article. The added content has been marked in red.

Thank you for your comments concerning our manuscript. Those comments are all valuable and very helpful for revising and improving our paper, as well as the important guiding significance to our researches. Once again, Thank you for your comments and valuable advice, and hope you can review it again in your busy schedule.

Reviewer 2 Report

In the present study the others reviewed the structural and bioacitive activities of daphnane-type diterpenoids classified into 5 types:6-epoxy daphnane diterpenoids, resiniferonoids, genkwanines, 1-Alkyldaphnanes and rediocides. This is a very interesting paper, the other hand this work presents some corrections according to the following comments.

Comments to Authors:

1.       The form of the 5 tables and the different chemical structures Daphnane-type Diterpenoids is poorly presented. Either you integrate the figure in the table must their corresponding names or created a figure including the chemical structures and their specifications.

2.       The manuscript includes a lot of detail and information, in order to be able to read and understand the paper well, I recommend to add another table and to include in it the source of each Daphnane-type Diterpenoids and especially their brief biological properties.

3.       Make some revision to the language sentences; the authors use long sentences, which make reading the paper difficult. Lighten up a little.

4.       It is interesting to update the reference list and citations should be update to include relevant recent works.

Author Response

Response to Reviewer 2 Comments

Point 1: The form of the 5 tables and the different chemical structures Daphnane-type Diterpenoids is poorly presented. Either you integrate the figure in the table must their corresponding names or created a figure including the chemical structures and their specifications.

Response 1: After careful consideration of the Reviewer’s suggestion. We've seen that each type of structure is too complex to be classified as one carbon skeleton. We have tried our best to integrate the chemical structures and their specifications, and have reduced it to the most simplified level.

Point 2: The manuscript includes a lot of detail and information, in order to be able to read and understand the paper well, I recommend to add another table and to include in it the source of each Daphnane-type Diterpenoids and especially their brief biological properties.

Response 2: We've added a table 1, and including the source of each daphnane-type diterpenoids and especially their medicinal active sites.

Point 3: Make some revision to the language sentences; the authors use long sentences, which make reading the paper difficult. Lighten up a little.

Response 3: We are very regretful for our negligence of the details. We have rewritten some long sentences into short sentences to make them easier to understand.

Point 4: It is interesting to update the reference list and citations should be update to include relevant recent works.

Response 4: I have read and cited recent works and updated the references.

Thank you for your comments. I am very sorry that I have been delayed upload the manuscript for a few days because I accompanied my parents on the International Labor Day holiday. We have studied comments carefully and have made correction which we hope meet with appral. Revised portion are marked in red in the paper. Special thanks to you for your good commets and suggestions, and hope that the correction will meet with approval.

Round 2

Reviewer 2 Report

The authors have made all the corrections and suggestions that have been requested, I recommend the publication of this manuscript.